# Improved Dynamic Regret for Non-degenerate Functions

**Lijun Zhang**[*], **Tianbao Yang**[†], **Jinfeng Yi**[‡], **Rong Jin**[§], **Zhi-Hua Zhou**[*]
[*]National Key Laboratory for Novel Software Technology, Nanjing University, Nanjing, China
[†]Department of Computer Science, The University of Iowa, Iowa City, USA
[‡]AI Foundations Lab, IBM Thomas J. Watson Research Center, Yorktown Heights, NY, USA
[§]Alibaba Group, Seattle, USA
`zhanglj@lamda.nju.edu.cn, tianbao-yang@uiowa.edu, jinfengyi@tencent.com`
`jinrong.jr@alibaba-inc.com, zhouzh@lamda.nju.edu.cn`

## Abstract

Recently, there has been a growing research interest in the analysis of dynamic regret, which measures the performance of an online learner against a sequence of local minimizers. By exploiting the strong convexity, previous studies have shown that the dynamic regret can be upper bounded by the path-length of the comparator sequence. In this paper, we illustrate that the dynamic regret can be further improved by allowing the learner to query the gradient of the function multiple times, and meanwhile the strong convexity can be weakened to other non-degenerate conditions. Specifically, we introduce the *squared* path-length, which could be much smaller than the path-length, as a new regularity of the comparator sequence. When multiple gradients are accessible to the learner, we first demonstrate that the dynamic regret of strongly convex functions can be upper bounded by the minimum of the path-length and the squared path-length. We then extend our theoretical guarantee to functions that are semi-strongly convex or self-concordant. To the best of our knowledge, this is the first time that semi-strong convexity and self-concordance are utilized to tighten the dynamic regret.

## 1 Introduction

Online convex optimization is a fundamental tool for solving a wide variety of machine learning problems [Shalev-Shwartz, 2011]. It can be formulated as a repeated game between a learner and an adversary. On the $t$-th round of the game, the learner selects a point $\mathbf{x}_t$ from a convex set $\mathcal{X}$ and the adversary chooses a convex function $f_t : \mathcal{X} \mapsto \mathbb{R}$. Then, the function is revealed to the learner, who incurs loss $f_t(\mathbf{x}_t)$. The standard performance measure is the regret, defined as the difference between the learner's cumulative loss and the cumulative loss of the optimal fixed vector in hindsight:

$$\sum_{t=1}^{T} f_t(\mathbf{x}_t) - \min_{\mathbf{x} \in \mathcal{X}} \sum_{t=1}^{T} f_t(\mathbf{x}). \tag{1}$$

Over the past decades, various online algorithms, such as the online gradient descent [Zinkevich, 2003], have been proposed to yield sub-linear regret under different scenarios [Hazan et al., 2007, Shalev-Shwartz et al., 2007].

Though equipped with rich theories, the notion of regret fails to illustrate the performance of online algorithms in dynamic setting, as a *static* comparator is used in (1). To overcome this limitation, there has been a recent surge of interest in analyzing a more stringent metric—*dynamic* regret [Hall and Willett, 2013, Besbes et al., 2015, Jadbabaie et al., 2015, Mokhtari et al., 2016, Yang et al.,

2016], in which the cumulative loss of the learner is compared against a sequence of local minimizers, i.e.,

$$R_T^* := R(\mathbf{x}_1^*, \ldots, \mathbf{x}_T^*) = \sum_{t=1}^{T} f_t(\mathbf{x}_t) - \sum_{t=1}^{T} f_t(\mathbf{x}_t^*) = \sum_{t=1}^{T} f_t(\mathbf{x}_t) - \sum_{t=1}^{T} \min_{\mathbf{x} \in \mathcal{X}} f_t(\mathbf{x}) \qquad (2)$$

where $\mathbf{x}_t^* \in \operatorname{argmin}_{\mathbf{x} \in \mathcal{X}} f_t(\mathbf{x})$. A more general definition of dynamic regret is to evaluate the difference of the cumulative loss with respect to any sequence of comparators $\mathbf{u}_1, \ldots, \mathbf{u}_T \in \mathcal{X}$ [Zinkevich, 2003].

It is well-known that in the worst-case, it is impossible to achieve a sub-linear dynamic regret bound, due to the arbitrary fluctuation in the functions. However, it is possible to upper bound the dynamic regret in terms of certain regularity of the comparator sequence or the function sequence. A natural regularity is the path-length of the comparator sequence, defined as

$$\mathcal{P}_T^* := \mathcal{P}(\mathbf{x}_1^*, \ldots, \mathbf{x}_T^*) = \sum_{t=2}^{T} \|\mathbf{x}_t^* - \mathbf{x}_{t-1}^*\| \qquad (3)$$

that captures the cumulative Euclidean norm of the difference between successive comparators. For convex functions, the dynamic regret of online gradient descent can be upper bounded by $O(\sqrt{T}\mathcal{P}_T^*)$ [Zinkevich, 2003]. And when all the functions are strongly convex and smooth, the upper bound can be improved to $O(\mathcal{P}_T^*)$ [Mokhtari et al., 2016].

In the aforementioned results, the learner uses the gradient of each function only *once*, and performs one step of gradient descent to update the intermediate solution. In this paper, we examine an interesting question: is it possible to improve the dynamic regret when the learner is allowed to query the gradient *multiple* times? Note that the answer to this question is no if one aims to promote the static regret in (1), according to the results on the minimax regret bound [Abernethy et al., 2008a]. We however show that when coming to the dynamic regret, multiple gradients can reduce the upper bound significantly. To this end, we introduce a new regularity—the *squared* path-length:

$$\mathcal{S}_T^* := \mathcal{S}(\mathbf{x}_1^*, \ldots, \mathbf{x}_T^*) = \sum_{t=2}^{T} \|\mathbf{x}_t^* - \mathbf{x}_{t-1}^*\|^2 \qquad (4)$$

which could be much smaller than $\mathcal{P}_T^*$ when the local variations are small. For example, when $\|\mathbf{x}_t^* - \mathbf{x}_{t-1}^*\| = \Omega(1/\sqrt{T})$ for all $t \in [T]$, we have $\mathcal{P}_T^* = \Omega(\sqrt{T})$ but $\mathcal{S}_T^* = \Omega(1)$. We advance the analysis of dynamic regret in the following aspects.

- When all the functions are strongly convex and smooth, we propose to apply gradient descent multiple times in each round, and demonstrate that the dynamic regret is reduced from $O(\mathcal{P}_T^*)$ to $O(\min(\mathcal{P}_T^*, \mathcal{S}_T^*))$, provided the gradients of minimizers are small. We further present a matching lower bound which implies our result cannot be improved in general.
- When all the functions are semi-strongly convex and smooth, we show that the standard online gradient descent still achieves the $O(\mathcal{P}_T^*)$ dynamic regret. And if we apply gradient descent multiple times in each round, the upper bound can also be improved to $O(\min(\mathcal{P}_T^*, \mathcal{S}_T^*))$, under the same condition as strongly convex functions.
- When all the functions are self-concordant, we establish a similar guarantee if both the gradient and Hessian of the function can be queried multiple times. Specifically, we propose to apply the damped Newton method multiple times in each round, and prove an $O(\min(\mathcal{P}_T^*, \mathcal{S}_T^*))$ bound of the dynamic regret under appropriate conditions.[1]

**Application to Statistical Learning**  Most studies of dynamic regret, including this paper do not make stochastic assumptions on the function sequence. In the following, we discuss how to interpret our results when facing the problem of statistical learning. In this case, the learner receives a sequence of losses $\ell(\mathbf{x}^\top \mathbf{z}_1, y_1), \ell(\mathbf{x}^\top \mathbf{z}_2, y_2), \ldots$, where $(\mathbf{z}_i, y_i)$'s are instance-label pairs sampled from a unknown distribution, and $\ell(\cdot, \cdot)$ measures the prediction error. To avoid the random fluctuation caused by sampling, we can set $f_t$ as the loss averaged over a mini-batch of instance-label pairs. As a result, when the underlying distribution is stationary or drifts slowly, successive functions will be close to each other, and thus the path-length and the squared path-length are expected to be small.

## 2 Related Work

The static regret in (1) has been extensively studied in the literature [Shalev-Shwartz, 2011]. It has been established that the static regret can be upper bounded by $O(\sqrt{T})$, $O(\log T)$, and $O(\log T)$ for convex functions, strongly convex functions, and exponentially concave functions, respectively [Zinkevich, 2003, Hazan et al., 2007]. Furthermore, those upper bounds are proved to be minimax optimal [Abernethy et al., 2008a, Hazan and Kale, 2011].

The notion of dynamic regret is introduced by Zinkevich [2003]. If we choose the online gradient descent as the learner, the dynamic regret with respect to any comparator sequence $\mathbf{u}_1, \ldots, \mathbf{u}_T$, i.e., $R(\mathbf{u}_1, \ldots, \mathbf{u}_T)$, is on the order of $\sqrt{T}\mathcal{P}(\mathbf{u}_1, \ldots, \mathbf{u}_T)$. When a prior knowledge of $\mathcal{P}_T^*$ is available, the dynamic regret $R_T^*$ can be upper bounded by $O(\sqrt{T\mathcal{P}_T^*})$ [Yang et al., 2016]. If all the functions are strongly convex and smooth, the upper bound of $R_T^*$ can be improved to $O(\mathcal{P}_T^*)$ [Mokhtari et al., 2016]. The $O(\mathcal{P}_T^*)$ rate is also achievable when all the functions are convex and smooth, and all the minimizers $\mathbf{x}_t^*$'s lie in the interior of $\mathcal{X}$ [Yang et al., 2016].

Another regularity of the comparator sequence, which is similar to the path-length, is defined as

$$\mathcal{P}'(\mathbf{u}_1, \ldots, \mathbf{u}_T) = \sum_{t=2}^{T} \|\mathbf{u}_t - \Phi_t(\mathbf{u}_{t-1})\|$$

where $\Phi_t(\cdot)$ is a dynamic model that predicts a reference point for the $t$-th round. The advantage of this measure is that when the comparator sequence follows the dynamical model closely, it can be much smaller than the path-length $\mathcal{P}(\mathbf{u}_1, \ldots, \mathbf{u}_T)$. A novel algorithm named dynamic mirror descent is proposed to take $\Phi_t(\mathbf{u}_{t-1})$ into account, and the dynamic regret $R(\mathbf{u}_1, \ldots, \mathbf{u}_T)$ is on the order of $\sqrt{T}\mathcal{P}'(\mathbf{u}_1, \ldots, \mathbf{u}_T)$ [Hall and Willett, 2013]. There are also some regularities defined in terms of the function sequence, such as the functional variation [Besbes et al., 2015]

$$\mathcal{F}_T := \mathcal{F}(f_1, \ldots, f_T) = \sum_{t=2}^{T} \max_{\mathbf{x} \in \mathcal{X}} |f_t(\mathbf{x}) - f_{t-1}(\mathbf{x})| \tag{5}$$

or the gradient variation [Chiang et al., 2012]

$$\mathcal{G}_T := \mathcal{G}(f_1, \ldots, f_T) = \sum_{t=2}^{T} \max_{\mathbf{x} \in \mathcal{X}} \|\nabla f_t(\mathbf{x}) - \nabla f_{t-1}(\mathbf{x})\|^2. \tag{6}$$

Under the condition that $\mathcal{F}_T \leq F_T$ and $F_t$ is given beforehand, a restarted online gradient descent is developed by Besbes et al. [2015], and the dynamic regret is upper bounded by $O(T^{2/3} F_T^{1/3})$ and $O(\log T \sqrt{TF_T})$ for convex functions and strongly convex functions, respectively.

The regularities mentioned above reflect different aspects of the learning problem, and are not directly comparable in general. Thus, it is appealing to develop an algorithm that adapts to the smaller regularity of the problem. Jadbabaie et al. [2015] propose an adaptive algorithm based on the optimistic mirror descent [Rakhlin and Sridharan, 2013], such that the dynamic regret is given in terms of all the three regularities ($\mathcal{P}_T^*$, $\mathcal{F}_T$, and $\mathcal{G}_T$). However, it relies on the assumption that the learner can calculate each regularity incrementally.

In the setting of prediction with expert advice, the dynamic regret is also referred to as tracking regret or shifting regret [Herbster and Warmuth, 1998, Cesa-bianchi et al., 2012]. The path-length of the comparator sequence is named as shift, which is just the number of times the expert changes. Another related performance measure is the adaptive regret, which aims to minimize the static regret over any interval [Hazan and Seshadhri, 2007, Daniely et al., 2015]. Finally, we note that the study of dynamic regret is similar to the competitive analysis in the sense that both of them compete against an optimal offline policy, but with significant differences in their assumptions and techniques [Buchbinder et al., 2012].

## 3 Online Learning with Multiple Gradients

In this section, we discuss how to improve the dynamic regret by allowing the learner to query the gradient multiple times. We start with strongly convex functions, and then proceed to semi-strongly convex functions, and finally investigate self-concordant functions.

**Algorithm 1** Online Multiple Gradient Descent (OMGD)

---
**Require:** The number of inner iterations $K$ and the step size $\eta$
1: Let $\mathbf{x}_1$ be any point in $\mathcal{X}$
2: **for** $t = 1, \ldots, T$ **do**
3:    Submit $\mathbf{x}_t \in \mathcal{X}$ and the receive loss $f_t : \mathcal{X} \mapsto \mathbb{R}$
4:    $\mathbf{z}_t^1 = \mathbf{x}_t$
5:    **for** $j = 1, \ldots, K$ **do**
6:

$$\mathbf{z}_t^{j+1} = \Pi_{\mathcal{X}} \left( \mathbf{z}_t^j - \eta \nabla f_t(\mathbf{z}_t^j) \right)$$

7:    **end for**
8:    $\mathbf{x}_{t+1} = \mathbf{z}_t^{K+1}$
9: **end for**

---

### 3.1 Strongly Convex and Smooth Functions

To be self-contained, we provide the definitions of strong convexity and smoothness.

**Definition 1.** *A function $f : \mathcal{X} \mapsto \mathbb{R}$ is $\lambda$-strongly convex, if*

$$f(\mathbf{y}) \geq f(\mathbf{x}) + \langle \nabla f(\mathbf{x}), \mathbf{y} - \mathbf{x} \rangle + \frac{\lambda}{2} \|\mathbf{y} - \mathbf{x}\|^2, \ \forall \mathbf{x}, \mathbf{y} \in \mathcal{X}.$$

**Definition 2.** *A function $f : \mathcal{X} \mapsto \mathbb{R}$ is $L$-smooth, if*

$$f(\mathbf{y}) \leq f(\mathbf{x}) + \langle \nabla f(\mathbf{x}), \mathbf{y} - \mathbf{x} \rangle + \frac{L}{2} \|\mathbf{y} - \mathbf{x}\|^2, \ \forall \mathbf{x}, \mathbf{y} \in \mathcal{X}.$$

**Example 1.** *The following functions are both strongly convex and smooth.*

    *1. A quadratic form $f(\mathbf{x}) = \mathbf{x}^\top A \mathbf{x} - 2\mathbf{b}^\top \mathbf{x} + c$ where $aI \preceq \mathcal{A} \preceq bI$, $a > 0$ and $b < \infty$;*
    *2. The regularized logistic loss $f(\mathbf{x}) = \log(1 + \exp(\mathbf{b}^\top \mathbf{x})) + \frac{\lambda}{2} \|\mathbf{x}\|^2$, where $\lambda > 0$.*

Following previous studies [Mokhtari et al., 2016], we make the following assumptions.

**Assumption 1.** *Suppose the following conditions hold for each $f_t : \mathcal{X} \mapsto \mathbb{R}$.*

    *1. $f_t$ is $\lambda$-strongly convex and $L$-smooth over $\mathcal{X}$;*
    *2. $\|\nabla f_t(\mathbf{x})\| \leq G, \ \forall \mathbf{x} \in \mathcal{X}$.*

When the learner can query the gradient of each function only once, the most popular learning algorithm is the online gradient descent:

$$\mathbf{x}_{t+1} = \Pi_{\mathcal{X}} \left( \mathbf{x}_t - \eta \nabla f_t(\mathbf{x}_t) \right)$$

where $\Pi_{\mathcal{X}}(\cdot)$ denotes the projection onto the nearest point in $\mathcal{X}$. Mokhtari et al. [2016] have established an $O(\mathcal{P}_T^*)$ bound of dynamic regret, as stated below.

**Theorem 1.** *Suppose Assumption 1 is true. By setting $\eta \leq 1/L$ in online gradient descent, we have*

$$\sum_{t=1}^{T} f_t(\mathbf{x}_t) - f_t(\mathbf{x}_t^*) \leq \frac{1}{1 - \gamma} G \mathcal{P}_T^* + \frac{1}{1 - \gamma} G \|\mathbf{x}_1 - \mathbf{x}_1^*\|$$

*where $\gamma = \sqrt{1 - \frac{2\lambda}{1/\eta + \lambda}}$.*

We now consider the setting that the learner can access the gradient of each function multiple times. The algorithm is a natural extension of online gradient descent by performing gradient descent multiple times in each round. Specifically, in the $t$-th round, given the current solution $\mathbf{x}_t$, we generate a sequence of solutions, denoted by $\mathbf{z}_t^1, \ldots, \mathbf{z}_t^{K+1}$, where $K$ is a constant independent from $T$, as follows:

$$\mathbf{z}_t^1 = \mathbf{x}_t, \quad \mathbf{z}_t^{j+1} = \Pi_{\mathcal{X}} \left( \mathbf{z}_t^j - \eta \nabla f_t(\mathbf{z}_t^j) \right), \ j = 1, \ldots, K.$$

Then, we set $\mathbf{x}_{t+1} = \mathbf{z}_t^{K+1}$. The procedure is named as Online Multiple Gradient Descent (OMGD) and is summarized in Algorithm 1.

By applying gradient descent multiple times, we are able to extract more information from each function and therefore are more likely to obtain a tight bound for the dynamic regret. The following theorem shows that the multiple accesses of the gradient indeed help improve the dynamic regret.

**Theorem 2.** *Suppose Assumption 1 is true. By setting $\eta \leq 1/L$ and $K = \lceil \frac{1/\eta + \lambda}{2\lambda} \ln 4 \rceil$ in Algorithm 1, for any constant $\alpha > 0$, we have*

$$\sum_{t=1}^{T} f_t(\mathbf{x}_t) - f_t(\mathbf{x}_t^*) \leq \min \begin{cases} 2G\mathcal{P}_T^* + 2G\|\mathbf{x}_1 - \mathbf{x}_1^*\|, \\ \frac{\sum_{t=1}^{T} \|\nabla f_t(\mathbf{x}_t^*)\|^2}{2\alpha} + 2(L+\alpha)\mathcal{S}_T^* + (L+\alpha)\|\mathbf{x}_1 - \mathbf{x}_1^*\|^2. \end{cases}$$

When $\sum_{t=1}^{T} \|\nabla f_t(\mathbf{x}_t^*)\|^2$ is small, Theorem 2 can be simplified as follows.

**Corollary 3.** *Suppose $\sum_{t=1}^{T} \|\nabla f_t(\mathbf{x}_t^*)\|^2 = O(\mathcal{S}_T^*)$, from Theorem 2, we have*

$$\sum_{t=1}^{T} f_t(\mathbf{x}_t) - f_t(\mathbf{x}_t^*) = O\left(\min(\mathcal{P}_T^*, \mathcal{S}_T^*)\right).$$

*In particular, if $\mathbf{x}_t^*$ belongs to the relative interior of $\mathcal{X}$ (i.e., $\nabla f_t(\mathbf{x}_t^*) = 0$) for all $t \in [T]$, Theorem 2, as $\alpha \to 0$, implies*

$$\sum_{t=1}^{T} f_t(\mathbf{x}_t) - f_t(\mathbf{x}_t^*) \leq \min\left(2G\mathcal{P}_T^* + 2G\|\mathbf{x}_1 - \mathbf{x}_1^*\|, 2L\mathcal{S}_T^* + L\|\mathbf{x}_1 - \mathbf{x}_1^*\|^2\right).$$

Compared to Theorem 1, the proposed OMGD improves the dynamic regret from $O(\mathcal{P}_T^*)$ to $O\left(\min\left(\mathcal{P}_T^*, \mathcal{S}_T^*\right)\right)$, when the gradients of minimizers are small. Recall the definitions of $\mathcal{P}_T^*$ and $\mathcal{S}_T^*$ in (3) and (4), respectively. We can see that $\mathcal{S}_T^*$ introduces a square when measuring the difference between $\mathbf{x}_t^*$ and $\mathbf{x}_{t-1}^*$. In this way, if the local variations ($\|\mathbf{x}_t^* - \mathbf{x}_{t-1}^*\|$'s) are small, $\mathcal{S}_T^*$ can be significantly smaller than $\mathcal{P}_T^*$, as indicated below.

**Example 2.** *Suppose $\|\mathbf{x}_t^* - \mathbf{x}_{t-1}^*\| = T^{-\tau}$ for all $t \geq 1$ and $\tau > 0$, we have*

$$\mathcal{S}_{T+1}^* = T^{1-2\tau} \ll \mathcal{P}_{T+1}^* = T^{1-\tau}.$$

*In particular, when $\tau = 1/2$, we have $\mathcal{S}_{T+1}^* = 1 \ll \mathcal{P}_{T+1}^* = \sqrt{T}$.*

$\mathcal{S}_T^*$ is also closely related to the gradient variation in (6). When all the $\mathbf{x}_t^*$'s belong to the relative interior of $\mathcal{X}$, we have $\nabla f_t(\mathbf{x}_t^*) = 0$ for all $t \in [T]$ and therefore

$$\mathcal{G}_T \geq \sum_{t=2}^{T} \|\nabla f_t(\mathbf{x}_{t-1}^*) - \nabla f_{t-1}(\mathbf{x}_{t-1}^*)\|^2 = \sum_{t=2}^{T} \|\nabla f_t(\mathbf{x}_{t-1}^*) - \nabla f_t(\mathbf{x}_t^*)\|^2 \geq \lambda^2 \mathcal{S}_T^* \quad (7)$$

where the last inequality follows from the property of strongly convex functions [Nesterov, 2004]. The following corollary is an immediate consequence of Theorem 2 and the inequality in (7).

**Corollary 4.** *Suppose Assumption 1 is true, and further assume all the $\mathbf{x}_t^*$'s belong to the relative interior of $\mathcal{X}$. By setting $\eta \leq 1/L$ and $K = \lceil \frac{1/\eta + \lambda}{2\lambda} \ln 4 \rceil$ in Algorithm 1, we have*

$$\sum_{t=1}^{T} f_t(\mathbf{x}_t) - f_t(\mathbf{x}_t^*) \leq \min\left(2G\mathcal{P}_T^* + 2G\|\mathbf{x}_1 - \mathbf{x}_1^*\|, \frac{2L\mathcal{G}_T}{\lambda^2} + L\|\mathbf{x}_1 - \mathbf{x}_1^*\|^2\right).$$

In Theorem 2, the number of accesses of gradients $K$ is set to be a constant depending on the condition number of the function. One may ask whether we can obtain a tighter bound by using a larger $K$. Unfortunately, according to our analysis, even if we take $K = \infty$, which means $f_t(\cdot)$ is minimized exactly, the upper bound can only be improved by a constant factor and the order remains the same. A related question is whether we can reduce the value of $K$ by adopting more advanced optimization techniques, such as the accelerated gradient descent [Nesterov, 2004]. This is an open problem to us, and will be investigated as a future work.

Finally, we prove that the $O(\mathcal{S}_T^*)$ bound is optimal for strongly convex and smooth functions.

**Theorem 5.** *For any online learning algorithm $\mathcal{A}$, there always exists a sequence of strongly convex and smooth functions $f_1, \ldots, f_T$, such that*

$$\sum_{t=1}^{T} f_t(\mathbf{x}_t) - f_t(\mathbf{x}_t^*) = \Omega(\mathcal{S}_T^*)$$

*where $\mathbf{x}_1, \ldots, \mathbf{x}_T$ is the solutions generated by $\mathcal{A}$.*

Thus, the upper bound in Theorem 2 cannot be improved in general.

### 3.2 Semi-strongly Convex and Smooth Functions

During the analysis of Theorems 1 and 2, we realize that the proof is built upon the fact that "when the function is strongly convex and smooth, gradient descent can reduce the distance to the optimal solution by a constant factor" [Mokhtari et al., 2016, Proposition 2]. From the recent developments in convex optimization, we know that a similar behavior also happens when the function is semi-strongly convex and smooth [Necoara et al., 2015, Theorem 5.2], which motivates the study in this section.

We first introduce the definition of semi-strong convexity [Gong and Ye, 2014].

**Definition 3.** *A function $f : \mathcal{X} \mapsto \mathbb{R}$ is semi-strongly convex over $\mathcal{X}$, if there exists a constant $\beta > 0$ such that for any $\mathbf{x} \in \mathcal{X}$*

$$f(\mathbf{x}) - \min_{\mathbf{x} \in \mathcal{X}} f(\mathbf{x}) \geq \frac{\beta}{2} \left\| \mathbf{x} - \Pi_{\mathcal{X}^*}(\mathbf{x}) \right\|^2 \tag{8}$$

*where $\mathcal{X}^* = \{\mathbf{x} \in \mathcal{X} : f(\mathbf{x}) \leq \min_{\mathbf{x} \in \mathcal{X}} f(\mathbf{x})\}$ is the set of minimizers of $f$ over $\mathcal{X}$.*

The semi-strong convexity generalizes several non-strongly convex conditions, such as the quadratic approximation property and the error bound property [Wang and Lin, 2014, Necoara et al., 2015]. A class of functions that satisfy the semi-strongly convexity is provided below [Gong and Ye, 2014].

**Example 3.** *Consider the following constrained optimization problem*

$$\min_{\mathbf{x} \in \mathcal{X} \subseteq \mathbb{R}^d} f(\mathbf{x}) = g(E\mathbf{x}) + \mathbf{b}^\top \mathbf{x}$$

*where $g(\cdot)$ is strongly convex and smooth, and $\mathcal{X}$ is either $\mathbb{R}^d$ or a polyhedral set. Then, $f : \mathcal{X} \mapsto \mathbb{R}$ is semi-strongly convex over $\mathcal{X}$ with some constant $\beta > 0$.*

Based on the semi-strong convexity, we assume the functions satisfy the following conditions.

**Assumption 2.** *Suppose the following conditions hold for each $f_t : \mathcal{X} \mapsto \mathbb{R}$.*

*1. $f_t$ is semi-strongly convex over $\mathcal{X}$ with parameter $\beta > 0$, and $L$-smooth;*
*2. $\|\nabla f_t(\mathbf{x})\| \leq G, \forall \mathbf{x} \in \mathcal{X}$.*

When the function is semi-strongly convex, the optimal solution may not be unique. Thus, we need to redefine $P_T^*$ and $\mathcal{S}_T^*$ to account for this freedom. We define

$$\mathcal{P}_T^* := \sum_{t=2}^{T} \max_{\mathbf{x} \in \mathcal{X}} \left\| \Pi_{\mathcal{X}_t^*}(\mathbf{x}) - \Pi_{\mathcal{X}_{t-1}^*}(\mathbf{x}) \right\|, \text{ and } \mathcal{S}_T^* := \sum_{t=2}^{T} \max_{\mathbf{x} \in \mathcal{X}} \left\| \Pi_{\mathcal{X}_t^*}(\mathbf{x}) - \Pi_{\mathcal{X}_{t-1}^*}(\mathbf{x}) \right\|^2$$

where $\mathcal{X}_t^* = \{\mathbf{x} \in \mathcal{X} : f_t(\mathbf{x}) \leq \min_{\mathbf{x} \in \mathcal{X}} f_t(\mathbf{x})\}$ is the set of minimizers of $f_t$ over $\mathcal{X}$.

In this case, we will use the standard online gradient descent when the learner can query the gradient only once, and apply the online multiple gradient descent (OMGD) in Algorithm 1, when the learner can access the gradient multiple times. Using similar analysis as Theorems 1 and 2, we obtain the following dynamic regret bounds for functions that are semi-strongly convex and smooth.

**Theorem 6.** *Suppose Assumption 2 is true. By setting $\eta \leq 1/L$ in online gradient descent, we have*

$$\sum_{t=1}^{T} f_t(\mathbf{x}_t) - \sum_{t=1}^{T} \min_{\mathbf{x} \in \mathcal{X}} f_t(\mathbf{x}) \leq \frac{G\mathcal{P}_T^*}{1 - \gamma} + \frac{G\|\mathbf{x}_1 - \bar{\mathbf{x}}_1\|}{1 - \gamma}$$

*where $\gamma = \sqrt{1 - \frac{\beta}{1/\eta + \beta}}$, and $\bar{\mathbf{x}}_1 = \Pi_{\mathcal{X}_1^*}(\mathbf{x}_1)$.*

Thus, online gradient descent still achieves an $O(\mathcal{P}_T^*)$ bound of the dynamic regret.

**Theorem 7.** *Suppose Assumption 2 is true. By setting $\eta \leq 1/L$ and $K = \lceil \frac{1/\eta + \beta}{\beta} \ln 4 \rceil$ in Algorithm 1, for any constant $\alpha > 0$, we have*

$$\sum_{t=1}^{T} f_t(\mathbf{x}_t) - \sum_{t=1}^{T} \min_{\mathbf{x} \in \mathcal{X}} f_t(\mathbf{x}) \leq \min \begin{cases} 2G\mathcal{P}_T^* + 2G\|\mathbf{x}_1 - \bar{\mathbf{x}}_1\| \\ \frac{G_T^*}{2\alpha} + 2(L+\alpha)\mathcal{S}_T^* + (L+\alpha)\|\mathbf{x}_1 - \bar{\mathbf{x}}_1\|^2 \end{cases}$$

*where $G_T^* = \max_{\{\mathbf{x}_t^* \in \mathcal{X}_t^*\}_{t=1}^T} \sum_{t=1}^{T} \|\nabla f_t(\mathbf{x}_t^*)\|^2$, and $\bar{\mathbf{x}}_1 = \Pi_{\mathcal{X}_1^*}(\mathbf{x}_1)$.*

Again, when the gradients of minimizers are small, in other words, $G_T^* = O(\mathcal{S}_T^*)$, the proposed OMGD improves the dynamic regret form $O(\mathcal{P}_T^*)$ to $O(\min(\mathcal{P}_T^*, \mathcal{S}_T^*))$.

### 3.3 Self-concordant Functions

We extend our previous results to self-concordant functions, which could be non-strongly convex and even non-smooth. Self-concordant functions play an important role in interior-point methods for solving convex optimization problems. We note that in the study of bandit linear optimization [Abernethy et al., 2008b], self-concordant functions have been used as barriers for constraints. However, to the best of our knowledge, this is the first time that losses themselves are self-concordant.

The definition of self-concordant functions is given below [Nemirovski, 2004].

**Definition 4.** *Let $\mathcal{X}$ be a nonempty open convex set in $\mathbb{R}^d$ and $f$ be a $C^3$ convex function defined on $\mathcal{X}$. $f$ is called self-concordant on $\mathcal{X}$, if it possesses the following two properties:*

1. *$f(\mathbf{x}_i) \to \infty$ along every sequence $\{\mathbf{x}_i \in \mathcal{X}\}$ converging, as $i \to \infty$, to a boundary point of $\mathcal{X}$;*
2. *$f$ satisfies the differential inequality*

$$|D^3 f(\mathbf{x})[\mathbf{h}, \mathbf{h}, \mathbf{h}]| \leq 2 \left( \mathbf{h}^\top \nabla^2 f(\mathbf{x}) \mathbf{h} \right)^{3/2}$$

*for all $\mathbf{x} \in \mathcal{X}$ and all $\mathbf{h} \in \mathbb{R}^d$, where*

$$D^3 f(x)[\mathbf{h}_1, \mathbf{h}_2, \mathbf{h}_3] = \frac{\partial^3}{\partial t_1 \partial t_2 \partial t_3}|_{t_1 = t_2 = t_3 = 0} f(\mathbf{x} + t_1 \mathbf{h}_1 + t_2 \mathbf{h}_2 + t_3 \mathbf{h}_3) .$$

**Example 4.** *We provide some examples of self-concordant functions below [Boyd and Vandenberghe, 2004, Nemirovski, 2004].*

1. *The function $f(x) = -\log x$ is self-concordant on $(0, \infty)$.*
2. *A convex quadratic form $f(\mathbf{x}) = \mathbf{x}^\top A\mathbf{x} - 2\mathbf{b}^\top \mathbf{x} + c$ where $A \in \mathbb{R}^{d \times d}$, $\mathbf{b} \in \mathbb{R}^d$, and $c \in \mathbb{R}$, is self-concordant on $\mathbb{R}^d$.*
3. *If $f : \mathbb{R}^d \mapsto \mathbb{R}$ is self-concordant, and $A \in \mathbb{R}^{d \times k}$, $\mathbf{b} \in \mathbb{R}^d$, then $f(A\mathbf{x} + \mathbf{b})$ is self-concordant.*

Using the concept of self-concordance, we make the following assumptions.

**Assumption 3.** *Suppose the following conditions hold for each $f_t : \mathcal{X}_t \mapsto \mathbb{R}$.*

1. *$f_t$ is self-concordant on domain $\mathcal{X}_t$;*
2. *$f_t$ is non-degenerate on $\mathcal{X}_t$, i.e., $\nabla^2 f_t(\mathbf{x}) \succ 0, \forall x \in \mathcal{X}_t$;*
3. *$f_t$ attains its minimum on $\mathcal{X}_t$, and denote $\mathbf{x}_t^* = \operatorname{argmin}_{\mathbf{x} \in \mathcal{X}_t} f_t(\mathbf{x})$.*

Our approach is similar to previous cases except for the updating rule of $\mathbf{x}_t$. Since we do not assume functions are strongly convex, we need to take into account the second order structure when updating the current solution $\mathbf{x}_t$. Thus, we assume the learner can query both the gradient and Hessian of each function multiple times. Specifically, we apply the damped Newton method [Nemirovski, 2004] to update $\mathbf{x}_t$, as follows:

$$\mathbf{z}_t^1 = \mathbf{x}_t, \quad \mathbf{z}_t^{j+1} = \mathbf{z}_t^j - \frac{1}{1 + \lambda_t(\mathbf{z}_t^j)} \left[ \nabla^2 f_t(\mathbf{z}_t^j) \right]^{-1} \nabla f_t(\mathbf{z}_t^j), \ j = 1, \ldots, K$$

where

$$\lambda_t(\mathbf{z}_t^j) = \sqrt{\nabla f_t(\mathbf{z}_t^j)^\top \left[ \nabla^2 f_t(\mathbf{z}_t^j) \right]^{-1} \nabla f_t(\mathbf{z}_t^j)}. \tag{9}$$

---

**Algorithm 2** Online Multiple Newton Update (OMNU)

---

**Require:** The number of inner iterations $K$ in each round
1: Let $\mathbf{x}_1$ be any point in $\mathcal{X}_1$
2: **for** $t = 1, \ldots, T$ **do**
3:   Submit $\mathbf{x}_t \in \mathcal{X}$ and the receive loss $f_t : \mathcal{X} \mapsto \mathbb{R}$
4:   $\mathbf{z}_t^1 = \mathbf{x}_t$
5:   **for** $j = 1, \ldots, K$ **do**
6:

$$\mathbf{z}_t^{j+1} = \mathbf{z}_t^j - \frac{1}{1 + \lambda_t(\mathbf{z}_t^j)} \left[ \nabla^2 f_t(\mathbf{z}_t^j) \right]^{-1} \nabla f_t(\mathbf{z}_t^j)$$

   where $\lambda_t(\mathbf{z}_t^j)$ is given in (9)
7:   **end for**
8:   $\mathbf{x}_{t+1} = \mathbf{z}_t^{K+1}$
9: **end for**

---

Then, we set $\mathbf{x}_{t+1} = \mathbf{z}_t^{K+1}$. Since the damped Newton method needs to calculate the inverse of the Hessian matrix, its complexity is higher than gradient descent. The procedure is named as Online Multiple Newton Update (OMNU) and is summarized in Algorithm 2.

To analyze the dynamic regret of OMNU, we redefine the two regularities $\mathcal{P}_T^*$ and $\mathcal{S}_T^*$ as follows:

$$\mathcal{P}_T^* := \sum_{t=2}^{T} \|\mathbf{x}_t^* - \mathbf{x}_{t-1}^*\|_t = \sum_{t=2}^{T} \sqrt{(\mathbf{x}_t^* - \mathbf{x}_{t-1}^*)^\top \nabla^2 f_t(\mathbf{x}_t^*)(\mathbf{x}_t^* - \mathbf{x}_{t-1}^*)}$$

$$\mathcal{S}_T^* := \sum_{t=2}^{T} \|\mathbf{x}_t^* - \mathbf{x}_{t-1}^*\|_t^2 = \sum_{t=2}^{T} (\mathbf{x}_t^* - \mathbf{x}_{t-1}^*)^\top \nabla^2 f_t(\mathbf{x}_t^*)(\mathbf{x}_t^* - \mathbf{x}_{t-1}^*)$$

where $\|\mathbf{h}\|_t = \sqrt{\mathbf{h}^\top \nabla^2 f_t(\mathbf{x}_t^*)\mathbf{h}}$. Compared to the definitions in (3) and (4), we introduce $\nabla^2 f_t(\mathbf{x}_t^*)$ when measuring the distance between $\mathbf{x}_t^*$ and $\mathbf{x}_{t-1}^*$. When functions are strongly convex and smooth, these definitions are equivalent up to constant factors. We then define a quantity to compare the second order structure of consecutive functions:

$$\mu = \max_{t=2,\ldots,T} \left\{ \lambda_{\max} \left( \left[ \nabla^2 f_{t-1}(\mathbf{x}_{t-1}^*) \right]^{-1/2} \nabla^2 f_t(\mathbf{x}_t^*) \left[ \nabla^2 f_{t-1}(\mathbf{x}_{t-1}^*) \right]^{-1/2} \right) \right\} \tag{10}$$

where $\lambda_{\max}(\cdot)$ computes the maximum eigenvalue of its argument. When all the functions are $\lambda$-strongly convex and $L$-smooth, $\mu \leq L/\lambda$. Then, we have the following theorem regarding the dynamic regret of the proposed OMNU algorithm.

**Theorem 8.** *Suppose Assumption 3 is true, and further assume*

$$\|\mathbf{x}_{t-1}^* - \mathbf{x}_t^*\|_t^2 \leq \frac{1}{144}, \; \forall t \geq 2. \tag{11}$$

*When $t = 1$, we choose $K = O(1)(f_1(\mathbf{x}_1) - f_1(\mathbf{x}_1^*) + \log\log\mu)$ in OMNU such that*

$$\|\mathbf{x}_2 - \mathbf{x}_1^*\|_1^2 \leq \frac{1}{144\mu}. \tag{12}$$

*For $t \geq 2$, we set $K = \lceil \log_4(16\mu) \rceil$ in OMNU, then*

$$\sum_{t=1}^{T} f_t(\mathbf{x}_t) - f_t(\mathbf{x}_t^*) \leq \min\left( \frac{1}{3}\mathcal{P}_T^*, 4\mathcal{S}_T^* \right) + f_1(\mathbf{x}_1) - f_1(\mathbf{x}_1^*) + \frac{1}{36}.$$

The above theorem again implies the dynamic regret can be upper bounded by $O(\min(\mathcal{P}_T^*, \mathcal{S}_T^*))$ when the learner can access the gradient and Hessian multiple times. From the first property of self-concordant functions in Definition 4, we know that $\mathbf{x}_t^*$ must lie in the interior of $\mathcal{X}_t$, and thus $\nabla f_t(\mathbf{x}_t^*) = 0$ for all $t \in [T]$. As a result, we do not need the additional assumption that the gradients of minimizers are small, which has been used before to simplify Theorems 2 and 7.

Compared to Theorems 2 and 7, Theorem 8 introduces an additional condition in (11). This condition is required to ensure that $\mathbf{x}_t$ lies in the feasible region of $f_t(\cdot)$, otherwise, $f_t(\mathbf{x}_t)$ can be infinity

and it is impossible to bound the dynamic regret. The multiple applications of damped Newton method can enforce $\mathbf{x}_t$ to be close to $\mathbf{x}_{t-1}^*$. Combined with (11), we conclude that $\mathbf{x}_t$ is also close to $\mathbf{x}_t^*$. Then, based on the property of the Dikin ellipsoid of self-concordant functions [Nemirovski, 2004], we can guarantee that $\mathbf{x}_t$ is feasible for $f_t(\cdot)$.

## 4 Conclusion and Future Work

In this paper, we discuss how to reduce the dynamic regret of online learning by allowing the learner to query the gradient/Hessian of each function multiple times. By applying gradient descent multiple times in each round, we show that the dynamic regret can be upper bounded by the minimum of the path-length and the squared path-length, when functions are strongly convex and smooth. We then extend this theoretical guarantee to functions that are semi-strongly convex and smooth. We finally demonstrate that for self-concordant functions, applying the damped Newton method multiple times achieves a similar result.

In the current study, we upper bound the dynamic regret in terms of the path-length or the squared path-length of the comparator sequence. As we mentioned before, there also exist some regularities defined in terms of the function sequence, e.g., the functional variation [Besbes et al., 2015]. In the future, we will investigate whether multiple accesses of gradient/Hessian can improve the dynamic regret when measured by certain regularities of the function sequence. Another future work is to extend our results to the more general dynamic regret

$$R(\mathbf{u}_1, \ldots, \mathbf{u}_T) = \sum_{t=1}^{T} f_t(\mathbf{x}_t) - \sum_{t=1}^{T} f_t(\mathbf{u}_t)$$

where $\mathbf{u}_1, \ldots, \mathbf{u}_T \in \mathcal{X}$ is an arbitrary sequence of comparators [Zinkevich, 2003].

**Acknowledgments**

This work was partially supported by the NSFC (61603177, 61333014), JiangsuSF (BK20160658), YESS (2017QNRC001), NSF (IIS-1545995), and the Collaborative Innovation Center of Novel Software Technology and Industrialization. Jinfeng Yi is now at Tencent AI Lab, Bellevue, WA, USA.

## Footnotes

[1] $\mathcal{P}_T^*$ and $\mathcal{S}_T^*$ are modified slightly when functions are semi-strongly convex or self-concordant.

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
