[Supplementary Material · Dynamic_Supp.pdf]

# Supplementary Material: Improved Dynamic Regret for Non-degenerate Functions

**Lijun Zhang**[*], **Tianbao Yang**[†], **Jinfeng Yi**[‡], **Rong Jin**[§], **Zhi-Hua Zhou**[*]

[*]National Key Laboratory for Novel Software Technology, Nanjing University, Nanjing, China
[†]Department of Computer Science, The University of Iowa, Iowa City, USA
[‡]AI Foundations Lab, IBM Thomas J. Watson Research Center, Yorktown Heights, NY, USA
[§]Alibaba Group, Seattle, USA
zhanglj@lamda.nju.edu.cn, tianbao-yang@uiowa.edu, jinfengyi@tencent.com
jinrong.jr@alibaba-inc.com, zhouzh@lamda.nju.edu.cn

## A   Proof of Theorem 1

For the sake of completeness, we include the proof of Theorem 1, which was proved by Mokhtari et al. [2016]. We need the following property of gradient descent.

**Lemma 1.** *Assume that $f : \mathcal{X} \mapsto \mathbb{R}$ is $\lambda$-strongly convex and $L$-smooth, and $\mathbf{x}_* = \operatorname{argmin}_{\mathbf{x} \in \mathcal{X}} f(\mathbf{x})$. Let $\mathbf{v} = \Pi_{\mathcal{X}}(\mathbf{u} - \eta \nabla f(\mathbf{u}))$, where $\eta \leq 1/L$. We have*

$$\|\mathbf{v} - \mathbf{x}_*\| \leq \sqrt{1 - \frac{2\lambda}{1/\eta + \lambda}} \|\mathbf{u} - \mathbf{x}_*\|.$$

The constant in the above lemma is better than that in Proposition 2 of Mokhtari et al. [2016].

Since $\|\nabla f_t(\mathbf{x})\| \leq G$ for any $t \in [T]$ and any $\mathbf{x} \in \mathcal{X}$, we have

$$\sum_{t=1}^{T} f_t(\mathbf{x}_t) - f_t(\mathbf{x}_t^*) \leq G \sum_{t=1}^{T} \|\mathbf{x}_t - \mathbf{x}_t^*\|. \tag{13}$$

We now proceed to bound $\sum_{t=1}^{T} \|\mathbf{x}_t - \mathbf{x}_t^*\|$. By the triangle inequality, we have

$$\sum_{t=1}^{T} \|\mathbf{x}_t - \mathbf{x}_t^*\| \leq \|\mathbf{x}_1 - \mathbf{x}_1^*\| + \sum_{t=2}^{T} \left( \|\mathbf{x}_t - \mathbf{x}_{t-1}^*\| + \|\mathbf{x}_{t-1}^* - \mathbf{x}_t^*\| \right). \tag{14}$$

Since

$$\mathbf{x}_t = \Pi_{\mathcal{X}} \left( \mathbf{x}_{t-1} - \eta \nabla f_{t-1}(\mathbf{x}_{t-1}) \right)$$

using Lemma 1, we have

$$\|\mathbf{x}_t - \mathbf{x}_{t-1}^*\| \leq \gamma \|\mathbf{x}_{t-1} - \mathbf{x}_{t-1}^*\|. \tag{15}$$

From (14) and (15), we have

$$\sum_{t=1}^{T} \|\mathbf{x}_t - \mathbf{x}_t^*\| \leq \|\mathbf{x}_1 - \mathbf{x}_1^*\| + \gamma \sum_{t=2}^{T} \|\mathbf{x}_{t-1} - \mathbf{x}_{t-1}^*\| + \mathcal{P}_T^* \leq \|\mathbf{x}_1 - \mathbf{x}_1^*\| + \gamma \sum_{t=1}^{T} \|\mathbf{x}_t - \mathbf{x}_t^*\| + \mathcal{P}_T^*$$

implying

$$\sum_{t=1}^{T} \|\mathbf{x}_t - \mathbf{x}_t^*\| \leq \frac{1}{1-\gamma} \mathcal{P}_T^* + \frac{1}{1-\gamma} \|\mathbf{x}_1 - \mathbf{x}_1^*\|. \tag{16}$$

We complete the proof by substituting (16) into (13).

## B   Proof of Lemma 1

We first introduce the following property of strongly convex functions [Hazan and Kale, 2011].

**Lemma 2.** *Assume that $f : \mathcal{X} \mapsto \mathbb{R}$ is $\lambda$-strongly convex, and $\mathbf{x}_* = \operatorname{argmin}_{\mathbf{x} \in \mathcal{X}} f(\mathbf{x})$. Then, we have*

$$f(\mathbf{x}) - f(\mathbf{x}_*) \geq \frac{\lambda}{2} \|\mathbf{x} - \mathbf{x}_*\|^2, \ \forall \mathbf{x} \in \mathcal{X}. \tag{17}$$

From the updating rule, we have

$$\mathbf{v} = \operatorname*{argmin}_{\mathbf{x} \in \mathcal{X}} \ f(\mathbf{u}) + \langle \nabla f(\mathbf{u}), \mathbf{x} - \mathbf{u} \rangle + \frac{1}{2\eta} \|\mathbf{x} - \mathbf{u}\|^2.$$

According to Lemma 2, we have

$$
\begin{aligned}
&f(\mathbf{u}) + \langle \nabla f(\mathbf{u}), \mathbf{v} - \mathbf{u} \rangle + \frac{1}{2\eta} \|\mathbf{v} - \mathbf{u}\|^2 \\
&\leq f(\mathbf{u}) + \langle \nabla f(\mathbf{u}), \mathbf{x}_* - \mathbf{u} \rangle + \frac{1}{2\eta} \|\mathbf{x}_* - \mathbf{u}\|^2 - \frac{1}{2\eta} \|\mathbf{v} - \mathbf{x}_*\|^2.
\end{aligned}
\tag{18}
$$

Since $f(\mathbf{x})$ is $\lambda$-strongly convex, we have

$$f(\mathbf{u}) + \langle \nabla f(\mathbf{u}), \mathbf{x}_* - \mathbf{u} \rangle \leq f(\mathbf{x}_*) - \frac{\lambda}{2} \|\mathbf{x}_* - \mathbf{u}\|^2. \tag{19}$$

On the other hand, the smoothness assumption implies

$$f(\mathbf{v}) \leq f(\mathbf{u}) + \langle \nabla f(\mathbf{u}), \mathbf{v} - \mathbf{u} \rangle + \frac{L}{2} \|\mathbf{v} - \mathbf{u}\|^2 \leq f(\mathbf{u}) + \langle \nabla f(\mathbf{u}), \mathbf{v} - \mathbf{u} \rangle + \frac{1}{2\eta} \|\mathbf{v} - \mathbf{u}\|^2. \tag{20}$$

Combining (18), (19), and (20), we obtain

$$f(\mathbf{v}) \leq f(\mathbf{x}_*) - \frac{\lambda}{2} \|\mathbf{x}_* - \mathbf{u}\|^2 + \frac{1}{2\eta} \|\mathbf{x}_* - \mathbf{u}\|^2 - \frac{1}{2\eta} \|\mathbf{v} - \mathbf{x}_*\|^2. \tag{21}$$

Applying Lemma 2 again, we have

$$f(\mathbf{v}) - f(\mathbf{x}_*) \geq \frac{\lambda}{2} \|\mathbf{v} - \mathbf{x}_*\|^2. \tag{22}$$

We complete the proof by substituting (22) into (21) and rearranging.

## C   Proof of Theorem 2

Since $f_t(\cdot)$ is $L$-smooth, we have

$$f_t(\mathbf{x}_t) - f_t(\mathbf{x}_t^*) \leq \langle \nabla f_t(\mathbf{x}_t^*), \mathbf{x}_t - \mathbf{x}_t^* \rangle + \frac{L}{2} \|\mathbf{x}_t - \mathbf{x}_t^*\|^2 \leq \|\nabla f_t(\mathbf{x}_t^*)\| \|\mathbf{x}_t - \mathbf{x}_t^*\| + \frac{L}{2} \|\mathbf{x}_t - \mathbf{x}_t^*\|^2.$$

Combining with the fact

$$\|\nabla f_t(\mathbf{x}_t^*)\| \|\mathbf{x}_t - \mathbf{x}_t^*\| \leq \frac{1}{2\alpha} \|\nabla f_t(\mathbf{x}_t^*)\|^2 + \frac{\alpha}{2} \|\mathbf{x}_t - \mathbf{x}_t^*\|^2$$

for any $\alpha > 0$, we obtain

$$f_t(\mathbf{x}_t) - f_t(\mathbf{x}_t^*) \leq \frac{1}{2\alpha} \|\nabla f_t(\mathbf{x}_t^*)\|^2 + \frac{L + \alpha}{2} \|\mathbf{x}_t - \mathbf{x}_t^*\|^2.$$

Summing the above inequality over $t = 1, \ldots, T$, we get

$$\sum_{t=1}^{T} f_t(\mathbf{x}_t) - f_t(\mathbf{x}_t^*) \leq \frac{1}{2\alpha} \sum_{t=1}^{T} \|\nabla f_t(\mathbf{x}_t^*)\|^2 + \frac{L + \alpha}{2} \sum_{t=1}^{T} \|\mathbf{x}_t - \mathbf{x}_t^*\|^2. \tag{23}$$

We now proceed to bound $\sum_{t=1}^{T} \|\mathbf{x}_t - \mathbf{x}_t^*\|^2$. We have

$$\sum_{t=1}^{T} \|\mathbf{x}_t - \mathbf{x}_t^*\|^2 \leq \|\mathbf{x}_1 - \mathbf{x}_1^*\|^2 + 2 \sum_{t=2}^{T} \left( \|\mathbf{x}_t - \mathbf{x}_{t-1}^*\|^2 + \|\mathbf{x}_{t-1}^* - \mathbf{x}_t^*\|^2 \right). \tag{24}$$

Recall the updating rule

$$\mathbf{z}_{t-1}^{j+1} = \Pi_{\mathcal{X}}\left(\mathbf{z}_{t-1}^j - \eta\nabla f_{t-1}(\mathbf{z}_{t-1}^j)\right), \ j = 1,\dots,K.$$

From Lemma 1, we have

$$\|\mathbf{z}_{t-1}^{j+1} - \mathbf{x}_{t-1}^*\|^2 \le \left(1 - \frac{2\lambda}{1/\eta + \lambda}\right)\|\mathbf{z}_{t-1}^j - \mathbf{x}_{t-1}^*\|^2$$

which implies

$$\|\mathbf{x}_t - \mathbf{x}_{t-1}^*\|^2 = \|\mathbf{z}_{t-1}^{K+1} - \mathbf{x}_{t-1}^*\|^2 \le \left(1 - \frac{2\lambda}{1/\eta + \lambda}\right)^K \|\mathbf{x}_{t-1} - \mathbf{x}_{t-1}^*\|^2 \le \frac{1}{4}\|\mathbf{x}_{t-1} - \mathbf{x}_{t-1}^*\|^2 \tag{25}$$

where we choose $K = \lceil\frac{1/\eta+\lambda}{2\lambda}\ln 4\rceil$ such that

$$\left(1 - \frac{2\lambda}{1/\eta + \lambda}\right)^K \le \exp\left(-\frac{2K\lambda}{1/\eta + \lambda}\right) \le \frac{1}{4}.$$

From (24) and (25), we have

$$\sum_{t=1}^T \|\mathbf{x}_t - \mathbf{x}_t^*\|^2 \le \|\mathbf{x}_1 - \mathbf{x}_1^*\|^2 + \frac{1}{2}\sum_{t=2}^T \|\mathbf{x}_{t-1} - \mathbf{x}_{t-1}^*\|^2 + 2\mathcal{S}_T^*$$

$$\le \|\mathbf{x}_1 - \mathbf{x}_1^*\|^2 + \frac{1}{2}\sum_{t=1}^T \|\mathbf{x}_t - \mathbf{x}_t^*\|^2 + 2\mathcal{S}_T^*$$

implying

$$\sum_{t=1}^T \|\mathbf{x}_t - \mathbf{x}_t^*\|^2 \le 4\mathcal{S}_T^* + 2\|\mathbf{x}_1 - \mathbf{x}_1^*\|^2.$$

Substituting the above inequality into (23), we have

$$\sum_{t=1}^T f_t(\mathbf{x}_t) - f_t(\mathbf{x}_t^*) \le \frac{1}{2\alpha}\sum_{t=1}^T \|\nabla f_t(\mathbf{x}_t^*)\|^2 + 2(L+\alpha)\mathcal{S}_T^* + (L+\alpha)\|\mathbf{x}_1 - \mathbf{x}_1^*\|^2$$

for all $\alpha \ge 0$. Finally, we show that the dynamic regret can still be upper bounded by $\mathcal{P}_T^*$. From the previous analysis, we have

$$\|\mathbf{x}_t - \mathbf{x}_{t-1}^*\|^2 \overset{(25)}{\le} \frac{1}{4}\|\mathbf{x}_{t-1} - \mathbf{x}_{t-1}^*\|^2 \Rightarrow \|\mathbf{x}_t - \mathbf{x}_{t-1}^*\| \le \frac{1}{2}\|\mathbf{x}_{t-1} - \mathbf{x}_{t-1}^*\|.$$

Then, we can set $\gamma = 1/2$ in Theorem 1 and obtain

$$\sum_{t=1}^T f_t(\mathbf{x}_t) - f_t(\mathbf{x}_t^*) \le 2G\mathcal{P}_T^* + 2G\|\mathbf{x}_1 - \mathbf{x}_1^*\|.$$

# D   Proof of Theorem 5

We will randomly generate a sequence of functions $f_t : \mathbb{R}^d \mapsto \mathbb{R}, t = 1,\dots,T$, where each $f_t(\cdot)$ is independently sampled from a distribution $\mathcal{P}$. For any deterministic algorithm $\mathcal{A}$, it generates a sequence of solutions $\mathbf{x}_t \in \mathcal{X}, t = 1,\dots,T$, we define the expected dynamic regret as

$$\mathrm{E}\left[R_T^*\right] = \mathrm{E}\left[\sum_{t=1}^T f_t(\mathbf{x}_t) - f_t(\mathbf{x}_t^*)\right].$$

We will show that there exists a distribution of strongly convex and smooth functions such that for any fixed algorithm $\mathcal{A}$, we have $\mathrm{E}[R_T^*] \ge \mathrm{E}[\mathcal{S}_T^*]$.

For each round $t$, we randomly sample a vector $\varepsilon_t \in \mathbb{R}^d$ from the Gaussian distribution $\mathcal{N}(0, I)$. Using $\varepsilon_t$, we create a function

$$f_t(\mathbf{x}) = 2 \left\| \mathbf{x} - \tau \varepsilon_t \right\|^2$$

which is both strongly convex and smooth. Notice that $\mathbf{x}_t$ is independent from $\varepsilon_t$, and thus we can bound the expected dynamic regret as follows:

$$\mathrm{E}\left[ R_T^* \right] = \sum_{t=1}^{T} \mathrm{E}\left[ f_t(\mathbf{x}_t) - f_t(\mathbf{x}_t^*) \right] = 2 \sum_{t=1}^{T} \mathrm{E}\left[ \|\mathbf{x}_t\|^2 + d\tau^2 \right] \geq 2dT\tau^2.$$

We furthermore bound $\mathcal{S}_T^*$ as follows

$$\mathrm{E}[\mathcal{S}_T^*] = \sum_{t=2}^{T} \mathrm{E}\left[ \|\varepsilon_t - \varepsilon_{t-1}\|^2 \tau^2 \right] = 2d(T-1)\tau^2.$$

Therefore, $\mathrm{E}[R_T^*] \geq \mathrm{E}[\mathcal{S}_T^*]$. Hence, for any given algorithm $\mathcal{A}$, there exists a sequence of functions $f_1, \ldots, f_T$, such that $\sum_{t=1}^{T} f_t(\mathbf{x}_t) - f_t(\mathbf{x}_t^*) = \Omega(\mathcal{S}_T^*)$.

# E   Proof of Theorem 6

The proof is similar to that of Theorem 1.

We need the following property of gradient descent when applied to semi-strongly convex and smooth functions [Necoara et al., 2015], which is analogous to Lemma 1 developed for strongly convex functions.

**Lemma 3.** *Assume that $f(\cdot)$ is L-smooth and satisfies the semi-strong convexity condition in (8). Let $\mathbf{v} = \Pi_{\mathcal{X}}(\mathbf{u} - \eta \nabla f(\mathbf{u}))$, where $\eta \leq 1/L$. We have*

$$\left\| \mathbf{v} - \Pi_{\mathcal{X}^*}(\mathbf{v}) \right\| \leq \sqrt{1 - \frac{\beta}{1/\eta + \beta}} \left\| \mathbf{u} - \Pi_{\mathcal{X}_*}(\mathbf{u}) \right\|.$$

Since $\|\nabla f_t(\mathbf{x})\| \leq G$ for any $t \in [T]$ and any $\mathbf{x} \in \mathcal{X}$, we have

$$\sum_{t=1}^{T} f_t(\mathbf{x}_t) - \sum_{t=1}^{T} \min_{\mathbf{x} \in \mathcal{X}} f_t(\mathbf{x}) = \sum_{t=1}^{T} f_t(\mathbf{x}_t) - f_t \left( \Pi_{\mathcal{X}_t^*}(\mathbf{x}_t) \right) \leq G \sum_{t=1}^{T} \left\| \mathbf{x}_t - \Pi_{\mathcal{X}_t^*}(\mathbf{x}_t) \right\|. \qquad (26)$$

We now proceed to bound $\sum_{t=1}^{T} \|\mathbf{x}_t - \Pi_{\mathcal{X}_t^*}(\mathbf{x}_t)\|$. By the triangle inequality, we have

$$\sum_{t=1}^{T} \left\| \mathbf{x}_t - \Pi_{\mathcal{X}_t^*}(\mathbf{x}_t) \right\| \leq \left\| \mathbf{x}_1 - \Pi_{\mathcal{X}_1^*}(\mathbf{x}_1) \right\| + \sum_{t=2}^{T} \left( \left\| \mathbf{x}_t - \Pi_{\mathcal{X}_{t-1}^*}(\mathbf{x}_t) \right\| + \left\| \Pi_{\mathcal{X}_{t-1}^*}(\mathbf{x}_t) - \Pi_{\mathcal{X}_t^*}(\mathbf{x}_t) \right\| \right).$$

$$(27)$$

Since

$$\mathbf{x}_t = \Pi_{\mathcal{X}} \left( \mathbf{x}_{t-1} - \eta \nabla f_{t-1}(\mathbf{x}_{t-1}) \right)$$

using Lemma 3, we have

$$\left\| \mathbf{x}_t - \Pi_{\mathcal{X}_{t-1}^*}(\mathbf{x}_t) \right\| \leq \gamma \left\| \mathbf{x}_{t-1} - \Pi_{\mathcal{X}_{t-1}^*}(\mathbf{x}_{t-1}) \right\|. \qquad (28)$$

From (27) and (28), we have

$$\sum_{t=1}^{T} \left\| \mathbf{x}_t - \Pi_{\mathcal{X}_t^*}(\mathbf{x}_t) \right\|$$

$$\leq \left\| \mathbf{x}_1 - \Pi_{\mathcal{X}_1^*}(\mathbf{x}_1) \right\| + \gamma \sum_{t=2}^{T} \left\| \mathbf{x}_{t-1} - \Pi_{\mathcal{X}_{t-1}^*}(\mathbf{x}_{t-1}) \right\| + \sum_{t=2}^{T} \left\| \Pi_{\mathcal{X}_{t-1}^*}(\mathbf{x}_t) - \Pi_{\mathcal{X}_t^*}(\mathbf{x}_t) \right\|$$

$$\leq \left\| \mathbf{x}_1 - \Pi_{\mathcal{X}_1^*}(\mathbf{x}_1) \right\| + \gamma \sum_{t=1}^{T} \left\| \mathbf{x}_t - \Pi_{\mathcal{X}_t^*}(\mathbf{x}_t) \right\| + \mathcal{P}_T^*$$

implying

$$\sum_{t=1}^{T} \left\| \mathbf{x}_t - \Pi_{\mathcal{X}_t^*}(\mathbf{x}_t) \right\| \leq \frac{1}{1 - \gamma} \mathcal{P}_T^* + \frac{1}{1 - \gamma} \left\| \mathbf{x}_1 - \Pi_{\mathcal{X}_1^*}(\mathbf{x}_1) \right\|. \qquad (29)$$

We complete the proof by substituting (29) into (26).

# F   Proof of Lemma 3

For the sake of completeness, we provide the proof of Lemma 3, which can also be found in the work of Necoara et al. [2015].

The analysis is similar to that of Lemma 1. Define

$$\bar{\mathbf{u}} = \Pi_{\mathcal{X}^*}(\mathbf{u}), \text{ and } \bar{\mathbf{v}} = \Pi_{\mathcal{X}^*}(\mathbf{v}).$$

From the optimality condition of $\mathbf{v}$, we have

$$
\begin{aligned}
&f(\mathbf{u}) + \langle \nabla f(\mathbf{u}), \mathbf{v} - \mathbf{u} \rangle + \frac{1}{2\eta} \|\mathbf{v} - \mathbf{u}\|^2 \\
&\leq f(\mathbf{u}) + \langle \nabla f(\mathbf{u}), \bar{\mathbf{u}} - \mathbf{u} \rangle + \frac{1}{2\eta} \|\bar{\mathbf{u}} - \mathbf{u}\|^2 - \frac{1}{2\eta} \|\mathbf{v} - \bar{\mathbf{u}}\|^2.
\end{aligned}
\tag{30}
$$

From the convexity of $f(\mathbf{x})$, we have

$$f(\mathbf{u}) + \langle \nabla f(\mathbf{u}), \bar{\mathbf{u}} - \mathbf{u} \rangle \leq f(\bar{\mathbf{u}}). \tag{31}$$

Combining (30), (31), and (20), we obtain

$$f(\mathbf{v}) \leq f(\bar{\mathbf{u}}) + \frac{1}{2\eta} \|\bar{\mathbf{u}} - \mathbf{u}\|^2 - \frac{1}{2\eta} \|\mathbf{v} - \bar{\mathbf{u}}\|^2. \tag{32}$$

From the semi-strong convexity of $f(\cdot)$, we further have

$$f(\mathbf{v}) - f(\bar{\mathbf{u}}) \geq \frac{\beta}{2} \|\mathbf{v} - \bar{\mathbf{v}}\|^2.$$

Substituting the above inequality into (32), we have

$$\frac{1}{2\eta} \|\bar{\mathbf{u}} - \mathbf{u}\|^2 \geq \frac{1}{2\eta} \|\mathbf{v} - \bar{\mathbf{u}}\|^2 + \frac{\beta}{2} \|\mathbf{v} - \bar{\mathbf{v}}\|^2 \geq \left( \frac{1}{2\eta} + \frac{\beta}{2} \right) \|\mathbf{v} - \bar{\mathbf{v}}\|^2$$

which completes the proof.

# G   Proof of Theorem 7

The proof is similar to that of Theorem 2. In the following, we just provide the key differences.

Following the derivation of (23), we get

$$
\begin{aligned}
\sum_{t=1}^{T} f_t(\mathbf{x}_t) - \sum_{t=1}^{T} \min_{\mathbf{x} \in \mathcal{X}} f_t(\mathbf{x}) &\leq \frac{1}{2\alpha} \sum_{t=1}^{T} \left\| \nabla f_t \left( \Pi_{\mathcal{X}_t^*}(\mathbf{x}_t) \right) \right\|^2 + \frac{L+\alpha}{2} \sum_{t=1}^{T} \left\| \mathbf{x}_t - \Pi_{\mathcal{X}_t^*}(\mathbf{x}_t) \right\|^2 \\
&\leq \frac{1}{2\alpha} G_T^* + \frac{L+\alpha}{2} \sum_{t=1}^{T} \left\| \mathbf{x}_t - \Pi_{\mathcal{X}_t^*}(\mathbf{x}_t) \right\|^2
\end{aligned}
\tag{33}
$$

for any $\alpha > 0$.

To bound $\sum_{t=1}^{T} \|\mathbf{x}_t - \Pi_{\mathcal{X}_t^*}(\mathbf{x}_t)\|^2$, we have

$$\sum_{t=1}^{T} \left\| \mathbf{x}_t - \Pi_{\mathcal{X}_t^*}(\mathbf{x}_t) \right\|^2 \leq \left\| \mathbf{x}_1 - \Pi_{\mathcal{X}_1^*}(\mathbf{x}_1) \right\|^2 + 2 \sum_{t=2}^{T} \left( \left\| \mathbf{x}_t - \Pi_{\mathcal{X}_{t-1}^*}(\mathbf{x}_t) \right\|^2 + \left\| \Pi_{\mathcal{X}_{t-1}^*}(\mathbf{x}_t) - \Pi_{\mathcal{X}_t^*}(\mathbf{x}_t) \right\|^2 \right). \tag{34}$$

From Lemma 3 and the updating rule

$$\mathbf{z}_{t-1}^{j+1} = \Pi_{\mathcal{X}} \left( \mathbf{z}_{t-1}^{j} - \eta \nabla f_{t-1}(\mathbf{z}_{t-1}^{j}) \right), \quad j = 1, \ldots, K$$

we have

$$\left\| \mathbf{z}_{t-1}^{j+1} - \Pi_{\mathcal{X}_{t-1}^*}(\mathbf{z}_{t-1}^{j+1}) \right\|^2 \leq \left( 1 - \frac{\beta}{1/\eta + \beta} \right) \left\| \mathbf{z}_{t-1}^{j} - \Pi_{\mathcal{X}_{t-1}^*}(\mathbf{z}_{t-1}^{j}) \right\|^2, \quad j = 1, \ldots, K$$

which implies

$$
\begin{aligned}
\left\| \mathbf{x}_t - \Pi_{\mathcal{X}_{t-1}^*}(\mathbf{x}_t) \right\|^2 &= \left\| \mathbf{z}_{t-1}^{K+1} - \Pi_{\mathcal{X}_{t-1}^*}(\mathbf{z}_{t-1}^{K+1}) \right\|^2 \\
&\leq \left( 1 - \frac{\beta}{1/\eta + \beta} \right)^K \left\| \mathbf{x}_{t-1} - \Pi_{\mathcal{X}_{t-1}^*}(\mathbf{x}_{t-1}) \right\|^2 \leq \frac{1}{4} \left\| \mathbf{x}_{t-1} - \Pi_{\mathcal{X}_{t-1}^*}(\mathbf{x}_{t-1}) \right\|^2
\end{aligned}
\tag{35}
$$

where we choose $K = \lceil \frac{1/\eta + \beta}{\beta} \ln 4 \rceil$ such that

$$
\left( 1 - \frac{\beta}{1/\eta + \beta} \right)^K \leq \exp\left( -\frac{K\beta}{1/\eta + \beta} \right) \leq \frac{1}{4}.
$$

From (34) and (35), we have

$$
\begin{aligned}
\sum_{t=1}^T \left\| \mathbf{x}_t - \Pi_{\mathcal{X}_t^*}(\mathbf{x}_t) \right\|^2 &\leq \left\| \mathbf{x}_1 - \Pi_{\mathcal{X}_1^*}(\mathbf{x}_1) \right\|^2 + \frac{1}{2} \sum_{t=2}^T \left\| \mathbf{x}_{t-1} - \Pi_{\mathcal{X}_{t-1}^*}(\mathbf{x}_{t-1}) \right\|^2 + 2\mathcal{S}_T^* \\
&\leq \left\| \mathbf{x}_1 - \Pi_{\mathcal{X}_1^*}(\mathbf{x}_1) \right\|^2 + \frac{1}{2} \sum_{t=1}^T \left\| \mathbf{x}_t - \Pi_{\mathcal{X}_t^*}(\mathbf{x}_t) \right\|^2 + 2\mathcal{S}_T^*
\end{aligned}
\tag{36}
$$

implying

$$
\sum_{t=1}^T \left\| \mathbf{x}_t - \Pi_{\mathcal{X}_t^*}(\mathbf{x}_t) \right\|^2 \leq 4\mathcal{S}_T^* + 2\left\| \mathbf{x}_1 - \Pi_{\mathcal{X}_1^*}(\mathbf{x}_1) \right\|^2.
$$

Substituting the above inequality into (33), we have

$$
\sum_{t=1}^T f_t(\mathbf{x}_t) - \sum_{t=1}^T \min_{\mathbf{x} \in \mathcal{X}} f_t(\mathbf{x}) \leq \frac{1}{2\alpha} G_T^* + 2(L+\alpha)\mathcal{S}_T^* + (L+\alpha)\left\| \mathbf{x}_1 - \Pi_{\mathcal{X}_1^*}(\mathbf{x}_1) \right\|^2, \ \forall \alpha \geq 0.
$$

Finally, we show that the dynamic regret can still be upper bounded by $\mathcal{P}_T^*$. From the previous analysis, we have

$$
\left\| \mathbf{x}_t - \Pi_{\mathcal{X}_{t-1}^*}(\mathbf{x}_t) \right\| \overset{(35)}{\leq} \frac{1}{2} \left\| \mathbf{x}_{t-1} - \Pi_{\mathcal{X}_{t-1}^*}(\mathbf{x}_{t-1}) \right\|.
$$

Then, we can set $\gamma = 1/2$ in Theorem 6 and obtain

$$
\sum_{t=1}^T f_t(\mathbf{x}_t) - \sum_{t=1}^T \min_{\mathbf{x} \in \mathcal{X}} f_t(\mathbf{x}) \leq 2G\mathcal{P}_T^* + 2G\left\| \mathbf{x}_1 - \Pi_{\mathcal{X}_1^*}(\mathbf{x}_1) \right\|.
$$

## H   Proof of Theorem 8

The inequality (12) follows directly from the result in Section 2.2.X.C of Nemirovski [2004]. To prove the rest of this theorem, we will use the following properties of self-concordant functions and the damped Newton method [Nemirovski, 2004].

**Lemma 4.** *Let $f(\mathbf{x})$ be a self-concordant function, and $\|\mathbf{h}\|_{\mathbf{x}} = \sqrt{\mathbf{h}^\top \nabla^2 f(\mathbf{x})\mathbf{h}}$. Then, all points within the Dikin ellipsoid $W_{\mathbf{x}}$ centered at $\mathbf{x}$, defined as $W_{\mathbf{x}} = \{\mathbf{x}' : \|\mathbf{x}' - \mathbf{x}\|_{\mathbf{x}} \leq 1\}$, share similar second order structure. More specifically, for a given point $\mathbf{x}$ and for any $\mathbf{h}$ with $\|\mathbf{h}\|_{\mathbf{x}} \leq 1$, we have*

$$
(1 - \|\mathbf{h}\|_{\mathbf{x}})^2 \nabla^2 f(\mathbf{x}) \preceq \nabla^2 f(\mathbf{x} + \mathbf{h}) \preceq \frac{\nabla^2 f(\mathbf{x})}{(1 - \|\mathbf{h}\|_{\mathbf{x}})^2}.
\tag{37}
$$

*Define $\mathbf{x}^* = \operatorname{argmin}_{\mathbf{x}} f(\mathbf{x})$. Then, we have*

$$
\|\mathbf{x} - \mathbf{x}^*\|_{\mathbf{x}^*} \leq \frac{\lambda(\mathbf{x})}{1 - \lambda(\mathbf{x})}
\tag{38}
$$

*where $\lambda(\mathbf{x}) = \sqrt{\mathbf{x}^\top \left[ \nabla^2 f(\mathbf{x}) \right]^{-1} \mathbf{x}}$.*

*Consider the the damped Newton method: $\mathbf{v} = \mathbf{u} - \frac{1}{1+\lambda(\mathbf{u})} \left[ \nabla^2 f(\mathbf{u}) \right]^{-1} \nabla f(\mathbf{u})$. Then, we have*

$$
\lambda(\mathbf{v}) \leq 2\lambda^2(\mathbf{u}).
\tag{39}
$$

We will also use the following inequality frequently

$$\|\mathbf{x}\|_t^2 = \mathbf{x}^\top \nabla^2 f_t(\mathbf{x}_t^*)\mathbf{x}$$

$$=\mathbf{x}^\top \left[\nabla^2 f_{t-1}(\mathbf{x}_{t-1}^*)\right]^{\frac{1}{2}} \left[\nabla^2 f_{t-1}(\mathbf{x}_{t-1}^*)\right]^{-\frac{1}{2}} \nabla^2 f_t(\mathbf{x}_t^*) \left[\nabla^2 f_{t-1}(\mathbf{x}_{t-1}^*)\right]^{-\frac{1}{2}} \left[\nabla^2 f_{t-1}(\mathbf{x}_{t-1}^*)\right]^{\frac{1}{2}} \mathbf{x}$$

$$\overset{(10)}{\leq} \mu \mathbf{x}^\top \nabla^2 f_{t-1}(\mathbf{x}_{t-1}^*)\mathbf{x} = \mu \|\mathbf{x}\|_{t-1}^2.$$

$$(40)$$

We will assume that for any $t \geq 2$,

$$\|\mathbf{x}_t - \mathbf{x}_t^*\|_t \leq \frac{1}{6} \tag{41}$$

which will be proved at the end of the analysis.

According to the Taylor's theorem, for any $t \geq 2$, we have

$$f_t(\mathbf{x}_t) - f_t(\mathbf{x}_t^*) = \frac{1}{2}(\mathbf{x}_t - \mathbf{x}_t^*)^\top \nabla^2 f_t(\xi_t)(\mathbf{x}_t - \mathbf{x}_t^*)$$

where $\xi_t$ is a point on the line segment between $\mathbf{x}_t$ and $\mathbf{x}_t^*$. Now, using the property of self-concordant functions, we have

$$\nabla^2 f_t(\xi_t) = \nabla^2 f_t(\mathbf{x}_t^* + \xi_t - \mathbf{x}_t^*) \overset{(37)}{\preceq} \frac{1}{(1 - \|\xi_t - \mathbf{x}_t^*\|_t)^2} \nabla^2 f_t(\mathbf{x}_t^*) \preceq \frac{1}{(1 - \|\mathbf{x}_t - \mathbf{x}_t^*\|_t)^2} \nabla^2 f_t(\mathbf{x}_t^*)$$

where we use the inequality in (41) to ensure $\|\mathbf{x}_t - \mathbf{x}_t^*\|_t \leq 1$. We thus have

$$f_t(\mathbf{x}_t) - f_t(\mathbf{x}_t^*) \leq \frac{\|\mathbf{x}_t - \mathbf{x}_t^*\|_t^2}{2(1 - \|\mathbf{x}_t - \mathbf{x}_t^*\|_t)^2} \overset{(41)}{\leq} \|\mathbf{x}_t - \mathbf{x}_t^*\|_t^2.$$

As a result

$$\sum_{t=1}^T f_t(\mathbf{x}_t) - f_t(\mathbf{x}_t^*) \leq f_1(\mathbf{x}_1) - f_1(\mathbf{x}_1^*) + \sum_{t=2}^T \|\mathbf{x}_t - \mathbf{x}_t^*\|_t^2. \tag{42}$$

We first bound the dynamic regret by $\mathcal{S}_T^*$. To this end, we have

$$\sum_{t=2}^T \|\mathbf{x}_t - \mathbf{x}_t^*\|_t^2 \leq \sum_{t=2}^T 2\left(\|\mathbf{x}_t - \mathbf{x}_{t-1}^*\|_t^2 + \|\mathbf{x}_t^* - \mathbf{x}_{t-1}^*\|_t^2\right) \overset{(40)}{\leq} 2\mu \sum_{t=2}^T \|\mathbf{x}_t - \mathbf{x}_{t-1}^*\|_{t-1}^2 + 2\mathcal{S}_T^*. \tag{43}$$

We proceed to bound $\sum_{t=2}^T \|\mathbf{x}_t - \mathbf{x}_{t-1}^*\|_{t-1}^2$. Since $\mathbf{x}_t$ is derived by applying the damped Newton method multiple times to the initial solution $\mathbf{x}_{t-1}$, we need to first bound $\lambda_{t-1}(\mathbf{x}_{t-1})$. To this end, we establish the following lemma.

**Lemma 5.** *Let $f(\mathbf{x})$ be a self-concordant function, and $\mathbf{x}^* = \arg\min_\mathbf{x} f(\mathbf{x})$. If $\|\mathbf{u} - \mathbf{x}^*\|_{\mathbf{x}^*} < 1/2$, we have*

$$\lambda(\mathbf{u}) \leq \frac{1}{1 - 2\|\mathbf{u} - \mathbf{x}^*\|_{\mathbf{x}^*}} \|\mathbf{u} - \mathbf{x}^*\|_{\mathbf{x}^*}.$$

The above lemma implies

$$\lambda_{t-1}(\mathbf{x}_{t-1}) \leq \frac{1}{1 - 2\|\mathbf{x}_{t-1} - \mathbf{x}_{t-1}^*\|_{t-1}} \|\mathbf{x}_{t-1} - \mathbf{x}_{t-1}^*\|_{t-1} \overset{(41)}{\leq} \min\left(\frac{3}{2}\|\mathbf{x}_{t-1} - \mathbf{x}_{t-1}^*\|_{t-1}, \frac{1}{4}\right).$$

$$(44)$$

Recall the updating rule

$$\mathbf{z}_{t-1}^{j+1} = \mathbf{z}_{t-1}^j - \frac{1}{1 + \lambda_{t-1}(\mathbf{z}_{t-1}^j)} \left[\nabla^2 f_{t-1}(\mathbf{z}_{t-1}^j)\right]^{-1} \nabla f_{t-1}(\mathbf{z}_{t-1}^j), \; j = 1, \ldots, K.$$

From Lemma 4, we have

$$\lambda_{t-1}(\mathbf{z}_{t-1}^{j+1}) \overset{(39)}{\leq} 2\lambda_{t-1}^2(\mathbf{z}_{t-1}^j), \; j = 1, \ldots, K.$$

Since $\lambda_{t-1}(\mathbf{z}_{t-1}^1) = \lambda_{t-1}(\mathbf{x}_{t-1}) \leq 1/4$. By induction, it is easy to verify

$$\lambda_{t-1}(\mathbf{z}_{t-1}^j) \leq \frac{1}{4}, \; j = 1, \ldots, K, K+1. \tag{45}$$

Therefore,

$$\lambda_{t-1}(\mathbf{x}_t) = \lambda_{t-1}(\mathbf{z}_{t-1}^{K+1}) \leq \frac{1}{2}\lambda_{t-1}(\mathbf{z}_{t-1}^K) \leq \cdots \leq \frac{1}{2^K}\lambda_{t-1}(\mathbf{z}_{t-1}^1) = \frac{1}{2^K}\lambda_{t-1}(\mathbf{x}_{t-1}). \tag{46}$$

Again, using Lemma 4, we have

$$\|\mathbf{x}_t - \mathbf{x}_{t-1}^*\|_{t-1} \overset{(38)}{\leq} \frac{\lambda_{t-1}(\mathbf{x}_t)}{1 - \lambda_{t-1}(\mathbf{x}_t)} \overset{(45),(46)}{\leq} \frac{4}{3}\frac{1}{2^K}\lambda_{t-1}(\mathbf{x}_{t-1}) \overset{(44)}{\leq} \frac{2}{2^K}\|\mathbf{x}_{t-1} - \mathbf{x}_{t-1}^*\|_{t-1}$$

implying

$$\|\mathbf{x}_t - \mathbf{x}_{t-1}^*\|_{t-1}^2 \leq \frac{4}{4^K}\|\mathbf{x}_{t-1} - \mathbf{x}_{t-1}^*\|_{t-1}^2. \tag{47}$$

Combining (43) with (47), we have

$$\sum_{t=2}^T \|\mathbf{x}_t - \mathbf{x}_t^*\|_t^2 \leq \frac{8\mu}{4^K}\sum_{t=3}^T \|\mathbf{x}_{t-1} - \mathbf{x}_{t-1}^*\|_{t-1}^2 + 2\mu\|\mathbf{x}_2 - \mathbf{x}_1^*\|_1^2 + 2\mathcal{S}_T^* \tag{48}$$

$$\leq \frac{1}{2}\sum_{t=2}^T \|\mathbf{x}_t - \mathbf{x}_t^*\|_t^2 + 2\mu\|\mathbf{x}_2 - \mathbf{x}_1^*\|_1^2 + 2\mathcal{S}_T^*$$

where we use the fact $\frac{8\mu}{4^K} \leq 1/2$. From (48), we have

$$\sum_{t=2}^T \|\mathbf{x}_t - \mathbf{x}_t^*\|_t^2 \leq 4\mu\|\mathbf{x}_2 - \mathbf{x}_1^*\|_1^2 + 4\mathcal{S}_T^* \overset{(12)}{\leq} \frac{1}{36} + 4\mathcal{S}_T^*. \tag{49}$$

Substituting (49) into (42), we obtain

$$\sum_{t=1}^T f_t(\mathbf{x}_t) - f_t(\mathbf{x}_t^*) \leq 4\mathcal{S}_T^* + f_1(\mathbf{x}_1) - f_1(\mathbf{x}_1^*) + \frac{1}{36}.$$

Next, we bound the dynamic regret by $\mathcal{P}_T^*$. From (41) and (42), we immediately have

$$\sum_{t=1}^T f_t(\mathbf{x}_t) - f_t(\mathbf{x}_t^*) \leq f_1(\mathbf{x}_1) - f_1(\mathbf{x}_1^*) + \frac{1}{6}\sum_{t=2}^T \|\mathbf{x}_t - \mathbf{x}_t^*\|_t. \tag{50}$$

To bound the last term, we have

$$\sum_{t=2}^T \|\mathbf{x}_t - \mathbf{x}_t^*\|_t \leq \sum_{t=2}^T \left( \|\mathbf{x}_t - \mathbf{x}_{t-1}^*\|_t + \|\mathbf{x}_t^* - \mathbf{x}_{t-1}^*\|_t \right)$$

$$\overset{(40)}{\leq} \sqrt{\mu}\sum_{t=3}^T \|\mathbf{x}_t - \mathbf{x}_{t-1}^*\|_{t-1} + \sqrt{\mu}\|\mathbf{x}_2 - \mathbf{x}_1^*\|_1 + \mathcal{P}_T^*$$

$$\overset{(47),(12)}{\leq} \sqrt{\frac{4\mu}{4^K}}\sum_{t=3}^T \|\mathbf{x}_{t-1} - \mathbf{x}_{t-1}^*\|_{t-1} + \frac{1}{12} + \mathcal{P}_T^*$$

$$\leq \frac{1}{2}\sum_{t=2}^T \|\mathbf{x}_t - \mathbf{x}_t^*\|_t + \frac{1}{12} + \mathcal{P}_T^*$$

which implies

$$\sum_{t=2}^T \|\mathbf{x}_t - \mathbf{x}_t^*\|_t \leq \frac{1}{6} + 2\mathcal{P}_T^*. \tag{51}$$

Combining (50) and (51), we have

$$\sum_{t=1}^{T} f_t(\mathbf{x}_t) - f_t(\mathbf{x}_t^*) \leq \frac{1}{3}\mathcal{P}_T^* + f_1(\mathbf{x}_1) - f_1(\mathbf{x}_1^*) + \frac{1}{36}.$$

Finally, we prove that the inequality in (41) holds. For $t = 2$, we have

$$\|\mathbf{x}_2 - \mathbf{x}_2^*\|_2^2 \leq 2\|\mathbf{x}_2 - \mathbf{x}_1^*\|_2^2 + 2\|\mathbf{x}_1^* - \mathbf{x}_2^*\|_2^2 \overset{(11),(40)}{\leq} 2\mu\|\mathbf{x}_2 - \mathbf{x}_1^*\|_1^2 + \frac{1}{72} \overset{(12)}{\leq} \frac{1}{36}.$$

Now, we suppose (41) is true for $t = 2, \ldots, k$. We show (41) holds for $t = k + 1$. We have

$$\|\mathbf{x}_{k+1} - \mathbf{x}_{k+1}^*\|_{k+1}^2 \leq 2\|\mathbf{x}_{k+1} - \mathbf{x}_k^*\|_{k+1}^2 + 2\|\mathbf{x}_k^* - \mathbf{x}_{k+1}^*\|_{k+1}^2$$

$$\overset{(11),(40)}{\leq} 2\mu\|\mathbf{x}_{k+1} - \mathbf{x}_k^*\|_k^2 + \frac{1}{72} \overset{(47)}{\leq} \frac{8\mu}{4^K}\|\mathbf{x}_k - \mathbf{x}_k^*\|_k^2 + \frac{1}{72} \leq \frac{1}{2}\|\mathbf{x}_k - \mathbf{x}_k^*\|_k^2 + \frac{1}{72} \leq \frac{1}{36}.$$

# I   Proof of Lemma 5

By the mean value theorem for vector-valued functions, we have

$$\nabla f(\mathbf{u}) = \nabla f(\mathbf{u}) - \nabla f(\mathbf{x}^*) = \int_0^1 \nabla^2 f\left(\mathbf{x}^* + \tau(\mathbf{u} - \mathbf{x}^*)\right)(\mathbf{u} - \mathbf{x}^*)\,\mathrm{d}\tau. \tag{52}$$

Define

$$g(\mathbf{x}) = \mathbf{x}^\top \left[\nabla^2 f(\mathbf{u})\right]^{-1} \mathbf{x}$$

which is a convex function of $\mathbf{x}$. Then, we have

$$\lambda^2(\mathbf{u}) = \left\langle \nabla f(\mathbf{u}), \left[\nabla^2 f(\mathbf{u})\right]^{-1} \nabla f(\mathbf{u}) \right\rangle = g\left(\nabla f(\mathbf{u})\right)$$

$$\overset{(52)}{=} g\left(\int_0^1 \nabla^2 f\left(\mathbf{x}^* + \tau(\mathbf{u} - \mathbf{x}^*)\right)(\mathbf{u} - \mathbf{x}^*)\,\mathrm{d}\tau\right) \leq \int_0^1 g\left(\nabla^2 f\left(\mathbf{x}^* + \tau(\mathbf{u} - \mathbf{x}^*)\right)(\mathbf{u} - \mathbf{x}^*)\right)\mathrm{d}\tau \tag{53}$$

where the last step follows from Jensen's inequality.

Define $\xi_\tau = \mathbf{x}^* + \tau(\mathbf{u} - \mathbf{x}^*)$ which lies in the line segment between $\mathbf{u}$ and $\mathbf{x}^*$. In the following, we will provide an upper bound for

$$g\left(\nabla^2 f(\xi_\tau)(\mathbf{u} - \mathbf{x}^*)\right) = (\mathbf{u} - \mathbf{x}^*)^\top \nabla^2 f(\xi_\tau)\left[\nabla^2 f(\mathbf{u})\right]^{-1} \nabla^2 f(\xi_\tau)(\mathbf{u} - \mathbf{x}^*).$$

Following Lemma 4, we have

$$\nabla^2 f(\xi_\tau) = \nabla^2 f(\mathbf{x}^* + \xi_\tau - \mathbf{x}^*) \overset{(37)}{\preceq} \frac{1}{(1 - \|\xi_\tau - \mathbf{x}^*\|_{\mathbf{x}^*})^2} \nabla^2 f(\mathbf{x}^*) \preceq \frac{1}{(1 - \|\mathbf{u} - \mathbf{x}^*\|_{\mathbf{x}^*})^2} \nabla^2 f(\mathbf{x}^*), \tag{54}$$

$$\|\mathbf{u} - \xi_\tau\|_{\xi_\tau}^2 \overset{(54)}{\leq} \frac{\|\mathbf{u} - \xi_\tau\|_{\mathbf{x}^*}^2}{(1 - \|\mathbf{u} - \mathbf{x}^*\|_{\mathbf{x}^*})^2} \leq \frac{\|\mathbf{u} - \mathbf{x}^*\|_{\mathbf{x}^*}^2}{(1 - \|\mathbf{u} - \mathbf{x}^*\|_{\mathbf{x}^*})^2} < 1, \tag{55}$$

$$\nabla^2 f(\mathbf{u}) = \nabla^2 f(\xi_\tau + \mathbf{u} - \xi_\tau) \overset{(37)}{\succeq} (1 - \|\mathbf{u} - \xi_\tau\|_{\xi_\tau})^2 \nabla^2 f(\xi_\tau) \overset{(55)}{\succeq} \left(\frac{1 - 2\|\mathbf{u} - \mathbf{x}^*\|_{\mathbf{x}^*}}{1 - \|\mathbf{u} - \mathbf{x}^*\|_{\mathbf{x}^*}}\right)^2 \nabla^2 f(\xi_\tau). \tag{56}$$

As a result

$$g\left(\nabla^2 f(\xi_\tau)(\mathbf{u} - \mathbf{x}^*)\right) \overset{(56)}{\leq} \left(\frac{1 - \|\mathbf{u} - \mathbf{x}^*\|_{\mathbf{x}^*}}{1 - 2\|\mathbf{u} - \mathbf{x}^*\|_{\mathbf{x}^*}}\right)^2 \left\langle (\mathbf{u} - \mathbf{x}^*), \nabla^2 f(\xi_\tau)(\mathbf{u} - \mathbf{x}^*) \right\rangle$$

$$\overset{(54)}{\leq} \frac{1}{(1 - 2\|\mathbf{u} - \mathbf{x}^*\|_{\mathbf{x}^*})^2} \|\mathbf{u} - \mathbf{x}^*\|_{\mathbf{x}^*}^2. \tag{57}$$

We complete the proof by substituting (57) into (53).