[Reviews · NeurIPS 2017]

Reviewer 1



Summary: This paper studies dynamic regret of online convex optimization and propose a new notion of regularity of the comparator sequence, i.e. the sum of the *square* distances of consecutive optimal points, which could potentially be much smaller than the sum of distances studied previously. Bounds that are essentially the minimum of these two regularities are shown to be achieved by a simple idea of doing multiple steps of gradient descent/damped Newton method at each time for (semi-)strongly convex functions and self-concordant functions. Major Comments: The main contribution of the paper is this little cute idea of doing multiple steps of gradient descent, which turns out to lead to nontrivial and better dynamic regret bounds. Despite its simplicity, I found the results interesting and I like the fact that the achieved bounds are actually the minimum of the new regularity and the one used before. Overall the paper is well written, except that the intuition on why this idea works seems to be missing. It's not clear to me why it leads to this specific kind of bounds. All proof are in the appendix and I did not verify them. Typos: 1. L164, even -> even if 2. L187, blow -> below

Reviewer 2



In this paper, the authors study the problem of minimizing the dynamic regret in online learning. First, they introduce squared path-length to measure the complexity of the comparator sequence. Then, they demonstrate that if multiple gradients are accessible to the learner, the dynamic regret of strongly convex functions can be upper bounded by the minimum of the path-length and the squared path-length. Finally, they extend their theoretical guarantees to functions that are semi-strongly convex or self-concordant. This is a theoretical paper for analyzing the dynamic regret. The main difference from previous work is that the learner is able to query the gradient multiple times. The authors prove that under this feedback model, dynamic regret could be upper bounded by the minimum of the path-length and the squared path-length, which is a significant improvement when the squared path-length is small. Strong Points: 1. A new performance measure is introduced to bound the dynamic regret. 2. When functions are strongly convex, the authors develop a new optimization algorithm and prove its dynamic regret is upper bounded by the minimum of the path-length and the squared path-length. 3. This is the first time that semi-strong convexity and self-concordance are utilized to tighten the dynamic regret. Suggestions/Questions: 1. It is better to provide some empirical studies to support the theoretical results. 2. For self-concordant functions, why do we need an additional condition in (11)? 3. Due to the matrix inverse, the complexity of online multiple Newton update (OMNU) is much higher than online multiple gradient descent (OMGD), which should be mentioned explicitly.

Reviewer 3



The authors study online convex optimization with dynamic regret. While typical bounds in this setting are in terms of the path length of actions, the authors show that by performing multiple gradient descent updates at each round, one can bound the regret by the minimum of the path length and the squared path length. This is demonstrated in the setting of smooth and strongly convex functions and for semi-strongly convex functions. The authors also present a similar result for self-concordant functions by designing an algorithm that performs multiple Newton updates at each step. This paper is relatively clear and well-written. However, there are a couple of issues that lead to my relatively low score. The first and most obvious issue is that the theoretical bounds in this paper do not improve upon existing results. For these types of algorithms, the authors usually either present compelling examples of experiments demonstrating that their method is meaningful and useful. However, this is not done. Another issue is that unlike most work in this setting, and in fact most of online convex optimization, the bounds presented in this paper seem to only apply against the sequence of per-step minimizers and not arbitrary sequences of actions. For instance, if the learner were to compare against the sequence of static comparators in the original Zinkevich algorithm, the bound would automatically reflect this. This is not possible with the OMGD algorithm. I view this as a severe limitation of this work, and I believe it is a point that should at the very least be discussed in the paper.